# Responsibility: A General Instance and Training Process-based Explainable AI Approach

## Abstract

Explainable Artificial Intelligence (XAI) methods focus on helping human users better understand the decision making of an AI agent. However, many modern XAI approaches are not actionable to end users, particularly those without prior AI or ML knowledge. In this paper, we formally define and extend an XAI approach called *Responsibility*, which identifies the most responsible training instance for a particular model decision based on observing the model's training process. This instance can then be presented as an explanation: "this is what the AI agent learned that led to that decision." We present experimental results across a number of domains and architectures, along with the results of a user study. Our results demonstrate that Responsibility can help improve the performance of both human end users and secondary ML models.

## 1 Introduction

Explainable Artificial Intelligence (XAI) refers to methods aimed at making AI systems more understandable to human users (Biran & Cotton, 2017). While many such XAI approaches have been developed, a majority of them make use of post-hoc analysis, attempting to induce an explanation from observing the behavior of a final, trained model (Adadi & Berrada, 2018; Das & Rad, 2020; Schwalbe & Finzel, 2024). More broadly, it is challenging to design XAI systems that provide explanations which are understandable to non-expert users (Mohseni et al., 2021) and those that address these issues are often limited to a single application domain such as image classification (Vilone & Longo, 2020; 2021). Thus, despite the large number of existing XAI approaches, we still require novel, general methods that are understandable to non-experts.

In this paper, we investigate an XAI approach termed *Responsibility*[1] that focuses on direct examination of the training process rather than on post-hoc induction, motivated by a desire to answer the question: "What did the AI agent/model learn that led to making that decision?". We focus in particular on actionable answers in terms of giving users information to help them solve tasks. To accomplish this, the approach detects the training example that is the most *responsible* for a model's decision at inference time. While there have been several prior methods attempting to identify the most influential training samples (Koh & Liang, 2017; Ghorbani & Zou, 2019), they typically attempt to identify objective truths of the training data, regardless of the final state of a model after a particular training process. Given the stochastic nature of model training, the most influential training samples may not necessarily be the most decisive. In comparison, Responsibility observes a particular training process to determine what instances impacted the model during training and the kinds of impact they had. Thus, the focus here is on the effects of a specific training run on a final model. This arguably allows the approach to provide a more specific answer to the aforementioned question than prior instance-based approaches.

In this work, we significantly expand on prior work that applied Responsibility for co-creative design using a Reinforcement Learning agent (Khadivpour & Guzdial, 2020), by generalizing the approach with a more formal definition. We demonstrate the ability to apply this formalization to a wider range of models and tasks including image classification, text classification and image generation. Additionally, we evaluate the

---

[1]Note that this is unrelated to and not to be confused with *responsible AI*, which provides guidance for the implementation of AI approaches in organizations with a focus on fairness, transparency, accountability, and safety (Arrieta et al., 2020)

approach via a user study over an image classification task. We find that Responsibility can improve the performance of human and secondary models compared to prior XAI approaches.

## 2 Related Work

Explainable AI (XAI) methods aim to make AI models, particularly deep neural networks (DNN), more transparent and understandable. A majority of XAI approaches however have been limited to a particular type of problem with a recent survey (Nauta et al., 2023) of 312 XAI articles finding that 63% focused on classification tasks. Responsibility differs from such approaches as it is generalizable to different types of deep learning problems.

A large amount of prior XAI work focuses on image classification and relies on saliency in images (Itti et al., 1998). Saliency refers to each pixel's unique quality in an image. Several gradient-based methods have been used for explaining the predictions of DNNs (Simonyan et al., 2014; Sundararajan et al., 2017; Shrikumar et al., 2017). In such methods, each gradient measures how much a change in each input feature would impact the model's predictions in a small neighborhood around the input. Like many XAI approaches, these derive an explanation from a model after it is trained. In comparison, we focus on observing the changes in the model during the training process. In focusing on training, Responsibility can be viewed as falling under instance-based or example-based XAI methods, which study explainability through the lens of the training data in ML contexts. 'Example' here refers to individual training datapoints. These methods offer explanations by tracing the decisions of a model back to influential examples. One common method is influence functions (Koh & Liang, 2017) which identify the training samples with the greatest influence on a model's output for a given input by simulating the effect of removing or altering a single training sample on the model's output. There are many such example-based methods (Yeh et al., 2018; Feldman & Zhang, 2020; Hara et al., 2019; Ghorbani & Zou, 2019; Ilyas et al., 2022) however they typically require training a large number of models to have reliable results, which can make them impractical for large-scale or complex models (Akyürek et al., 2022). Some methods, such as those providing approximate influence scores or gradient-based scoring (Pruthi et al., 2020; Chen et al., 2021; Schioppa et al., 2022; Park et al., 2023), offer computational benefits by avoiding the need to retrain models. These approximations can be less reliable, particularly when the relationship between training data and model predictions is complex and nonlinear (Basu et al., 2020). Such approaches attempt to identify something objectively true about the training set, in many cases approximating a leave-one-out training process, in which each instance of the training data is left out and the model retrained from scratch. Responsibility instead focuses only on the process by which the final weights of a specific model emerged, ignoring other alternative training possibilities.

Classic XAI approaches for DNNs include class activation maps (CAMs) (Zhou et al., 2016; Selvaraju et al., 2017) which highlight the discriminative regions of an image used by a convolutional network to predict the class of the image. Other popular XAI methods for visualizing feature interactions and importance include Local Interpretable Model-agnostic Explanations (LIME) (Ribeiro et al., 2016) which derive explanations by training interpretable, surrogate models on local perturbations of predictions made by more complex, black box models and SHapley Additive exPlanations (SHAP) (Lundberg & Lee, 2017) which leverage Shapley values to estimate the contribution made by features towards a certain prediction. We compare Responsibility to these approaches in our human subject study.

The methods discussed above have been primarily designed for classification tasks. Extending these to generative models can be challenging due to the absence of clear labels or definitive ground truth. Prior work has looked at data influence in generative models such as GANs (Terashita et al., 2021), VAEs (Kong & Chaudhuri, 2021), and diffusion models (Lin et al., 2024; Dai & Gifford, 2023; Georgiev et al., 2023; Zheng et al., 2023). Responsibility differs from these model-specific approaches in terms of being applicable to both discriminative and generative tasks.

## 3 Responsibility

In this section, we define the Responsibility approach, which aims to identify the most *responsible* training sample(s) for a particular decision made by a model. Responsibility is approximated as a quantifiable value

representing the influence of each training sample on the model's learned behavior. It consists of three steps: 1) Responsibility allocation, 2) activation identification and 3) Responsibility-activation mapping. We describe each of these below.

### 3.1 Responsibility Allocation

This first step tracks the influence of training samples on weight updates. During training, the change in each weight at index $i$, in each training step, is calculated as:

$$\Delta w_i = w_i' - w_i \tag{1}$$

where $w_i$ and $w_i'$ represent the weight at index $i$ before and after a training step respectively. Each weight $w_i$ is associated with a *responsibility array* $R$ with size equal to the total number of training samples. During training, each calculated weight change $\Delta w_{i,m}$ associated with a sample $m$, is recorded in $R$ at the corresponding index of $m$. This accumulation of $\Delta w_{i,m}$ values during training enables tracking the cumulative influence of each sample $m$ on each weight $w_i$. The update to $R$ for each weight $w_m$ is as follows:

$$R_i[m] \mathrel{+}= \Delta w_{i,m} \tag{2}$$

where $R$ is the responsibility array, $m$ is the sample index in the training data and $\Delta w_{i,m}$ is the contribution of sample with index $m$ to the change in weight $w_i$. When training ends, $R_i$ represents how much each sample influenced the changes in weight $w_i$ during training. To allocate $\Delta w_i$ to samples, we introduce a *proportional allocation* method which accounts for the fact that different samples within a batch have varying levels of influence on the weight change (Ruder, 2016). This method proportionally allocates Responsibility by utilizing the loss function since it provides a reasonable approximation of the influence of a sample within a batch by reflecting how well the model performs for each sample with a higher loss indicating that the sample will have a larger influence on subsequent weight changes. Thus, using the loss allows estimating each sample's relative influence on the overall weight change. For each sample $j$ in the batch, its influence on $\Delta w_i$ is calculated as follows:

$$\Delta w_{i,j} = \frac{L_j}{L_{\text{batch}}} \cdot \Delta w_i \tag{3}$$

where $L_j$ is the loss for the $j^{th}$ sample and $L_{\text{batch}}$ is the total loss for the batch. This proportionally divides $\Delta w_i$ among all samples in the batch based on their individual losses, assigning more Responsibility to samples with higher loss values. Each $\Delta w_{i,j}$ then represents the contribution of $j^{th}$ sample of the batch to the overall weight change. We add this $\Delta w_{i,j}$ value to the index $m$ of the responsibility array $R$ as follows:

$$m = t * b + j \tag{4}$$

$$R_i[m] \mathrel{+}= \Delta w_{i,j} \tag{5}$$

where $t$ is the training step, $b$ is the batch size and $m$ is the sample index in the training data.

#### 3.1.1 Responsibility Allocation Types

After training, the responsibility array $R_i$ represents how much each training sample influenced the changes in weight $w_i$ during training and can thus be used to identify the most influential samples for $w_i$. There are several ways to do this, each offering different perspectives to understand how training samples influence model weights. In this paper, we consider the following allocation types:

**Positive Allocation**: $\text{resp}_{i,pos} = \arg\max((R_i))$
**Negative Allocation**: $\text{resp}_{i,neg} = \arg\min((R_i))$
**Maximal Allocation**: $\text{resp}_{i,max} = \arg\max(\text{abs}(R_i))$
**Minimal Allocation**: $\text{resp}_{i,\min} = \arg\min(\text{abs}(R_i))$

Positive and negative allocations could be useful in tasks like classification to pinpoint helpful or harmful data points respectively. Alternatively, maximal allocation highlights the most influential samples overall, regardless of polarity, making it useful for identifying key drivers of model behavior. Minimal allocation, in contrast, may be applied to models like diffusion models to identify samples with negligible effects on noise prediction i.e., samples that share the structure found at a particular step of the diffusion process.

### 3.2 Activation Identification

This step identifies the most critical weights of the network during inference. We start with a set of weights $W$, which includes all weights in a particular network layer. The choice of layer is domain-specific and depends on the target problem. From $W$, the goal is to identify a subset $W'$ containing the most critical weights during inference, which may include single or multiple weight(s). $W'$ can be determined by inspecting the activation of the layer: $a = g(x \cdot W)$ where $x$ is the input, $W$ is the weight matrix, $g$ is the activation function and $a$ is the activation. The most critical weights can then be selected based on the following strategies, analogous to the Responsibility allocation types:

**Positive Activations:** $argmax(a) \rightarrow W_{pos}$
**Negative Activations:** $argmin(a) \rightarrow W_{neg}$
**Maximal Activations:** $argmax(abs(a)) \rightarrow W_{max}$
**Minimal Activations:** $argmin(abs(a)) \rightarrow W_{min}$

These enable analyzing the behavior of weights from various perspectives. Positive and negative activations identify weights that most positively and negatively contribute to the output respectively, which might be useful in classification tasks by helping identify training samples that led to correct/incorrect predictions. On the other hand, maximal and minimal activations can help identify weights with the largest and smallest influence on model behavior.

### 3.3 Responsibility-Activation Mapping

This combines the prior steps to approximate the training samples that had the most influence on the most critical weights. For each weight, the most responsible training sample is determined using one of the Responsibility allocation strategies. Then, one of the activation identification strategies is used to pinpoint the most critical weights based on their activation values. The two are linked to identify the most responsible samples $M$ for each weight $w$ within $W'$. This enables drawing direct connections between individual training samples and critical activations during inference.

## 4 Classification Experiments

Since our goal with Responsibility is to improve human understanding, we performed an experiment to approximate this process using a secondary model, with the added goal of demonstrating the generalizability of Responsibility by applying it to both image and text classification. The evaluation uses a classification model, the actor, along with a secondary model, the critic, that attempts to predict the binary performance (right or wrong) of the actor.[2] This critic thus acts as a proxy for a human trying to improve their understanding of the actor model. We can test the impact of including or excluding a variety of output XAI explanations as part of the critic's training data. If these explanations increase its accuracy, it can be seen as improving its "understanding" of the actor, and may suggest that it could also improve human understanding. This can be seen as a type of simulated user study (Nauta et al., 2023).

For evaluation, we split our datasets into training, validation, and test sets and train the actor using the training set but do not try to achieve the best possible accuracy. In fact, it is best if the actor has relatively low accuracy as long as it is consistent. This lets us best evaluate the impact of different XAI explanations on the critic's accuracy. Thus after training, there are samples in our validation set that the actor could not classify correctly. We refer to these as incorrect validation instances. For the critic, we would ideally like a balanced dataset, with an equal number of cases in which the actor is correct and incorrect. Thus, we collect all the incorrect validation instances and sample an equal number of correct validation instances. We do not sample based on the classes in the actor's classification problem, since we focus on the binary question for the critic of whether the actor is right or wrong. We call this new dataset the baseline critic training set. This baseline set allows us to determine the accuracy of the critic without any explanations. We can then employ various XAI approaches to produce visual explanations to augment the data of this training set. We concatenate each baseline training set instance with its associated XAI explanation. Notably, this only

---

[2]Note that despite the terminology, this is not an example of a reinforcement learning actor-critic method.

works when the XAI approach can output explanations appropriate as model inputs. But given our focus on instance-based approaches, this is trivial. If we find an XAI approach that improves critic accuracy, it may make it possible to improve the accuracy of the actor, representing a direct benefit to the original model.

We perform evaluation on two problem domains: image classification using CIFAR-10 and sentiment classification using PL04. We chose these datasets for the purposes of illustration and due to their common usage. For both tasks, we compare Responsibility to a nearest neighbor (NN) approach, while also comparing to influence functions (IF) (Koh & Liang, 2017) for image classification. We do not include an instance-based XAI baseline for sentiment classification as none have been applied to PL04 to the best of our knowledge. For computing Responsibility values, we use positive allocations and activations. We use the weights $W$ of the last layer in the network since this layer directly impacts the prediction. From $W$, we determine a subset of the most critical weights $W'$ as previously discussed. A potential concern for Responsibility is whether the most responsible training instances capture the reasoning of the model, or if they simply represent the most similar training instance. Thus we include the NN method for which, we consider each instance of the baseline critic training set, identify the actor model's predicted class for this instance, and find the training instance most similar to the given baseline critic instance using mean squared error. This evaluation lets us determine comparatively how helpful each approach is to the critic learning the behavior of the actor and approximates how much each XAI approach would help an end-user understand and predict model behavior.

## 4.1  CIFAR-10

CIFAR-10 (Krizhevsky et al., 2009) is a popular image classification problem and dataset. We split the dataset into 30K training, 20K validation and 10K test sets. For our actor, we used a 4-layer CNN with max pooling and dropout, followed by 2 fully connected layers. We use ReLU throughout with a final softmax layer. We train the model using crossentropy loss via an SGD optimizer, a learning rate of 1e-3 and a batch size of 1. We train a second, simple 1-layer CNN as a critic. We use the same setup as the actor, except we use the Adam optimizer with a learning rate scheduler decreasing from 1e-4 to 1e-5 and a batch size of 32.

Table 1 shows the performance metrics of our critics on the test set. The Responsibility critic outperforms the other approaches by roughly 3% in this domain. This demonstrates a small but clear improvement over NN, influence functions or no extra information (Baseline). Of particular interest is that the baseline without extra information outperformed NN in terms of accuracy and precision. This suggests that NN is actively harmful to the critic, which follows from the fact that it does not reflect any information from the actor model outside of the predicted class. We note that the actor's accuracy on the test set is only 60%. While low for CIFAR-10, we expect this is due to the reduced training data. Further, as stated before, it was not our goal to train a highly accurate model.

|           | Resp. | IF | NN | Baseline |
|-----------|-------|-------|-------|----------|
| Accuracy  | **0.641** | 0.612 | 0.582 | 0.586 |
| Precision | **0.656** | 0.620 | 0.574 | 0.585 |
| Recall    | 0.583 | 0.584 | **0.638** | 0.590 |
| F1 Score  | **0.619** | 0.602 | 0.604 | 0.588 |

Table 1: CIFAR-10 domain performance metrics

|           | Resp. | NN | Baseline |
|-----------|-------|-------|----------|
| Accuracy  | **0.747** | 0.626 | 0.686 |
| Precision | 0.762 | **0.804** | 0.722 |
| Recall    | **0.974** | 0.661 | 0.658 |
| F1 Score  | **0.855** | 0.725 | 0.689 |

Table 2: The PL04 domain performance metrics

## 4.2  PL04

To test the generalizeability of Responsibility, we performed an experiment using the PL04 dataset (Pang & Lee, 2004) which consists of 2000 movie reviews equally divided into positive and negative sentiment classes. We preprocessed the data by eliminating punctuation marks, non-characters, extra spaces, and single-lettered words followed by lowercasing and lemmatization. We split our data into 1700 training, 150 validation, and 150 test sets, drawing equally from both classes for all splits. We used a CNN-LSTM architecture for both the actor and the critic models based on Camacho-Collados & Pilehvar (2017). We again used a batch size of 1 to simplify identification of the most responsible training instance. For validation, we compared the performance to the original published results. We performed a 10-fold cross-validation and found that

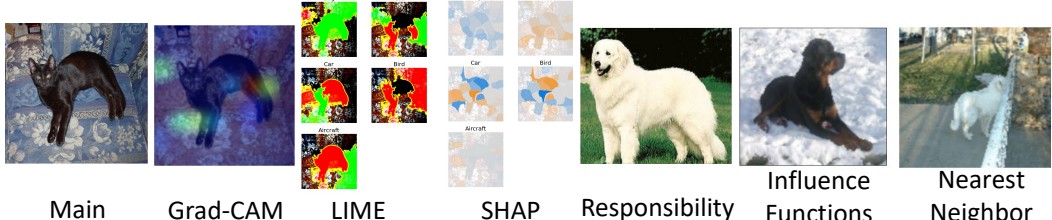

Figure 1: XAI approaches in the user study along with Responsibility and the Nearest Neighbor baseline.

we performed equivalently to the original work. We did not include influence functions as they were not originally designed to work with text data. Results are shown in Table 2. The Responsibility critic generally outperforms the others by roughly 10%. We again note the same effect from CIFAR-10 where NN led to worse accuracy. However, NN does outperform the other two approaches in precision. This may be due to there being only two classes, compared to the ten of CIFAR-10. The Responsibility critic is much more consistent, with a far higher recall. Since PL04 only has two classes, we can treat the accuracy of the critic as roughly equivalent to its accuracy on the actor's classification task. This means that the critic marginally improves over the performance of the actor (0.733 vs. 0.747). While this is a minor improvement, it demonstrates a further utility of Responsibility. Compared to CIFAR-10, PL04 seems better suited to Responsibility. However, this may be due to dataset size, models, and problem complexity (10 vs 2 classes). Thus, the approach is not necessarily more suited to NLP problems and our results mainly demonstrate that Responsibility can help an AI critic in multiple domains. To test if this generalizes to human users, we performed a human subject study.

## 5 Human Subject Study

We ran a user study[3] on Amazon Mechanical Turk (AMT) to compare Responsibility with prior XAI approaches. We chose AMT due to our focus on non-expert users and the number of subjects required given the size of this study. We focused on the task of users guessing the prediction of an image classification model which we chose as the majority of XAI work has focused on image classification (Nauta et al., 2023). We asked users to predict the classification of the model as an approximation for how well users understood the model's behavior from the XAI explanation. We used four popular XAI approaches for classification tasks: Local Interpretable Model-agnostic Explanations (LIME) (Ribeiro et al., 2016), Shapley Additive exPlanations (SHAP) (Lundberg & Lee, 2017), Grad-CAM (Selvaraju et al., 2017), and influence functions (IF) (Koh & Liang, 2017). We chose these as they are general approaches with distinct visualizations. We focused on visualizations so that non-experts could differentiate between the explanations. We also included the Nearest Neighbor (NN) approach from the prior experiment. This was a strong baseline, essentially informing the user of the correct class, but without any basis in what the model learned. For Responsibility, we again used the positive allocations and activations. Example visualizations of all approaches for a given test image are shown in Figure 1, with the input test image on the left. The correct class is cat but the predicted class is dog. The Responsibility image appears to clearly convey the predicted class, but may seem somewhat unusual or lacking similarity to the original test image. However, the dark background of the Responsibility image shares some similarity with the dark cat. Considering the LIME visualization, we can confirm that the model paid attention to the shared darkness between the two images. We found this type of shared characteristic common for the most responsible instances compared to particular input images.

### 5.1 Dataset and model

We used VGG-16 (Simonyan & Zisserman, 2014) as the trained model for comparing the explanations generated by the different methods. Unfortunately, many popular XAI image classification datasets have too low of a resolution for meaningful non-expert human judgment. We thus constructed a novel dataset

---

[3]This study was reviewed and approved by a Research Ethics Board at < Redacted for Anonymity >

|                    | Resp. | G-CAM | LIME | SHAP | IF   | NN   |
|--------------------|-------|-------|------|------|------|------|
| More Understandable | **59** | **59** | 50   | 36   | 46   | 50   |
| More Preferred      | 53    | **59** | 45   | 47   | 42   | 51   |
| Pair Accuracy       | 62.6  | 54.8  | 56.2 | 55.4 | 50.2 | **64.8** |
| Pref. Accuracy      | **83.6** | 47.5 | 50.0 | 53.3 | 42.8 | 81.5 |

Table 3: Overview of the AMT study results.

by combining 5 image datasets with relatively high resolutions—Caltech-UCSD Birds-200-2011 (Wah et al., 2011), CAT (Zhang et al., 2008), Stanford Dogs (Khosla et al., 2011), FGVC Aircraft (Maji et al., 2013), and Stanford Cars (Krause et al., 2013). We trained VGG-16 as a classifier on these 5 image classes, using the same training setup as in the CIFAR-10 evaluation in order to identify the responsible training instances. For this study, we only used test images and their associated explanations.

## 5.2 Methodology

We employed a within-subjects comparative evaluation due to the benefits of ranking evaluations in human subject studies in comparison to rating evaluations (Martinez et al., 2014). Thus, each user was randomly assigned two approaches. Since the users were non-experts, they were given a short tutorial on image classification and XAI. They were then introduced to their two assigned approaches with two example test images (one correct, one incorrect) and the associated explanations from both approaches. These introductory examples were fixed, so all users assigned to the same approaches saw the same images and visualization. For the main experiment, each user was given five images with the explanation visualizations of their two assigned approaches. For each image, users were informed of the correct class and the model's prediction. After these initial five "training" examples, they were given five new "test" images along with the associated explanation visualizations. However, for these five examples, users were not given the model's predicted class but instead asked to pick which of the five classes they thought the model predicted. The model's prediction could be correct or incorrect, and the users only had their experience and the pair of explanations to guide their decision. Thus the first five examples acted as a training set and the second five acted as a test set for the users. Finally, users were asked to rank the two explanations in terms of preference and understandability. Each user was guaranteed to see 10 unique test images, determined by their study ID, to minimize the risk of measuring differences between particular images instead of between the XAI approaches. This also ensured users could not collude on the correct model predictions. Each user also had an equal number of instances in which the model was correct and incorrect spread randomly across their examples. Users were given a maximum of 15 minutes, though all finished in under 10 minutes. Each user was paid 3 USD for participating and an additional 0.2 USD for each correct answer for the second five examples to incentivize them to guess correctly and to make use of the explanations.

## 5.3 Results

We collected results from 300 AMT workers, for a total of 3000 predictions, or 100 predictions for each approach. We ran a MANOVA to determine whether the ordering of the approaches or any of the other demographic information impacted any of the results and found no significant impact so we could safely treat this data as coming from only 15 conditions: the 15 distinct pairs of XAI approaches. We thus had 20 users in each condition. Results are shown in Table 3. "More Understandable" is the total number of times that particular approach was chosen as more understandable by one of the 75 users who saw it. We found that Responsibility and Grad-CAM were the two approaches most commonly chosen to be more understandable. Notably this is summing data over all possible pairs of approaches. A paired Wilcoxon Mann Whitney-U test, chosen due to a lack of a normal distribution, found that LIME was considered significantly more understandable than SHAP, and Responsibility was considered more understandable than LIME ($p < 0.005$). However, there was no significant difference between SHAP and Grad-CAM on understandability, suggesting that they were too similar for users to differentiate. "More Preferred" indicates the number of times, out of the 75 users who saw that approach, that the approach was overall preferred. Grad-CAM is the clear winner here with Responsibility coming second. There are no significant differences in terms of overall

preference. We also looked at "Pair Accuracy", the average accuracy taken across all five pairs that include each approach. We use this due to the complexity of determining whether the user's performance comes from one of the pair of explanations presented or from a combination of both. NN outperforms all other approaches according to this measure. This may seem trivial since NN explicitly informed the user of the model's chosen class. However, both Responsibility and influence functions also essentially informed the user of the model's chosen class (as in Figure 1). The only significant difference for this metric is that both Responsibility and NN have significantly different Pair Accuracy compared to influence functions. We also include "Pref. Accuracy" which is the average accuracy of the users when they indicated that the XAI approach in question was their preference ("More Preferred"). Here, Responsibility takes the lead, with NN performing very similarly, and all other approaches having a performance near chance. This may be due to users believing they understood an explanation per the "Most Understandable" results, but not being able to identify the predicted class from it. We note that influence functions should have performed well here but users seemed to have a harder time understanding how the selected image related to the test instance. Overall, these results suggest that Responsibility differs from existing XAI approaches, may be more beneficial to human understanding and further supports our prior experiment, demonstrating that improving critic accuracy does appear to approximate improving human accuracy.

# 6 Diffusion Model Experiment

To further demonstrate the generalizability of Responsibility outside of classification and to more recent models, we apply the approach to diffusion models, which are a powerful class of generative models that have shown remarkable performance in image generation tasks (Song et al., 2020; Nichol et al., 2021; Rombach et al., 2022). These models, introduced by Sohl-Dickstein et al. (2015), are inspired by physical diffusion where particles gradually spread from regions of high to low concentration. They operate by gradually refining random noise into coherent, high-quality samples. Denoising Diffusion Probabilistic Models (DDPM) (Ho et al., 2020) is one of the foundational types of diffusion models, based on a U-Net architecture (Ronneberger et al., 2015) which consists of an encoder and a decoder where the encoder gradually reduces the spatial dimensions and extracts features, and the decoder upsamples these features to reconstruct the image. Given the sequential nature of noise removal, our experiments focus on identifying the training samples most responsible for predicted noise at each step in the generation process.

## 6.1 Architecture

We use a modified version of the architecture of Ho et al. (2020) which in turn is based on the U-net architecture from Salimans et al. (2017). The model consists of multiple residual blocks, organized into down-sampling (encoder), middle, and up-sampling (decoder) stages with more than 63 million trainable parameters. The network accepts an image and a time step as inputs, with the time step guiding noise removal for each denoising step. Each residual block consists of 2 convolutional layers with $3 \times 3$ kernels, group normalization (Wu & He, 2018), and the swish activation function (Ramachandran et al., 2017), followed by a skip connection (He et al., 2016). Each attention block consists of a group normalization layer, followed by a dense layer to apply a self-attention mechanism (Vaswani, 2017).

## 6.2 Dataset

For our experiments, we used the CelebA (Liu et al., 2015) dataset which consists of over 200K celebrity face images and has been widely used in prior XAI research in diffusion models (Dai & Gifford, 2023; Zheng et al., 2023; Lin et al., 2024). We also used the Oxford Flowers 102 dataset (Nilsback & Zisserman, 2008) which contains 8,189 images across 102 flower categories and serves as a simpler domain for comparison, allowing us to evaluate the generalizability of our findings across datasets. For CelebA, to reduce computational cost, we used 32000 samples for training and for both datasets, we resized each image to $32 \times 32$ pixels.

### 6.3 Experiment Setup

When applied to diffusion models, Responsibility needs to be handled differently compared to classification or regression tasks. Since the network's output is noise to be subtracted from the input image, the output itself is effectively the opposite of the desired final image. Thus, we hypothesize that the training samples derived from minimal allocations may be more valuable compared with those derived from positive allocations, as they indicate the samples most responsible for improving the model's noise prediction accuracy. For our experiments, we focus on the weights $W$ of the last convolutional layer in the diffusion model as it directly impacts the network's output. From $W$, we identify a subset $W'$ that consists of the most critical weights.

Due to the lack of a ground truth, we use image similarity as an approximation for explanation quality, assessing how closely the most responsible samples resemble the generated image. Measuring the alignment between generated images and responsible samples serves as an approximation of humans assessing similarity based on shared visual features or structural patterns. We use two metrics: Structural Similarity Index (SSIM) and Mean Squared Error (MSE). SSIM (Wang et al., 2004) is a statistical metric that captures how structural patterns in one image align with those in another, making it useful for assessing generative models. SSIM values range from -1 to 1, with values closer to 1 indicating higher similarity. MSE is a commonly applied pixel-wise metric that focuses on individual pixels, by calculating the average squared difference between corresponding pixels of two images. Lower values reflect higher similarity, as they indicate greater alignment between the images. These metrics together provide a comprehensive assessment for whether the most responsible samples align closely with generated images or deviate as expected. For our experiments, we compared the Responsibility varieties against two baselines: nearest neighbor (NN) samples, derived as before, and samples derived via influence functions where we approximated the influence of training samples as outlined in (Charpiat et al., 2019). This calculates an influence score by determining the dot product between the gradients of a test sample and the training sample. We do not use the original influence function approximation (Koh & Liang, 2017) due to computational cost far exceeding the computation required for Responsibility. Since this method requires true values to calculate the loss and then compute gradients, we utilize test samples and reconstruct them using our diffusion model. By comparing these influential samples with the identified responsible samples, we evaluate which set is more similar to the reconstructed images. We focus on examining minimal and positive allocations. We reconstructed 20 randomly selected test samples. To evaluate the similarity of the identified most responsible training samples to generated images, we compare the Responsibility varieties against three baselines: nearest neighbor (NN) samples, random samples from the training set and the most influential samples derived from influence functions.

### 6.4 CelebA Results

In this section, we present the results of our CelebA experiments, focusing on the performance of our Responsibility variants. We examine minimal and positive allocations, which identify training samples that caused the smallest absolute changes and maximum positive changes in weights respectively. In both cases, we use different activation identification types to identify the responsible samples. Our focus is on these two allocations due to the interpretive value of these variants in the context of diffusion models. For completeness, we also computed all other Responsibility variants for the CelebA dataset with the results presented in the appendix. As anticipated, variants other than minimal and positive allocations showed poor performance. Thus, we limit our comparative analysis to these two allocation types in the subsequent sections.

#### 6.4.1 Minimal Allocations

Minimal allocations point to the samples that caused the smallest absolute changes in weights. We include all four activation identification types to identify $W'$ which can include single or multiple critical weights. The number of critical weights varies for each generated sample, as it depends on the number of outliers in the activation values. We calculate the similarity between our generated images and the responsible samples, compared to random samples from the training set. Table 4 summarizes the performance of each variant in terms of SSIM and MSE.

We observe that across almost all Responsibility types, responsible samples yield higher similarity to the generated images, as shown by the higher SSIM and lower MSE values for responsible samples compared to

| Activation type | SSIM | | | MSE | | |
|---|---|---|---|---|---|---|
| | **Responsibility** | **Random** | **NN** | **Responsibility** | **Random** | **NN** |
| Multi. maximal | $16.44 \pm 0.35$ | $13.91 \pm 0.12$ | $26.91 \pm 0.31$ | $14.95 \pm 0.12$ | $16.05 \pm 0.13$ | $5.51 \pm 0.02$ |
| Multi. minimal | $15.12 \pm 0.21$ | $13.96 \pm 0.14$ | $18.14 \pm 0.13$ | $15.08 \pm 0.07$ | $15.56 \pm 0.08$ | $9.33 \pm 0.03$ |
| Multi. negative | $13.88 \pm 0.14$ | $13.83 \pm 0.18$ | $27.73 \pm 0.33$ | $15.44 \pm 0.16$ | $15.33 \pm 0.17$ | $5.30 \pm 0.02$ |
| Multi. positive | $16.44 \pm 0.35$ | $14.15 \pm 0.18$ | $26.91 \pm 0.31$ | $14.95 \pm 0.12$ | $15.26 \pm 0.09$ | $5.51 \pm 0.02$ |
| Single maximal | $15.30 \pm 0.21$ | $14.06 \pm 0.15$ | $26.62 \pm 0.32$ | $14.90 \pm 0.14$ | $15.16 \pm 0.12$ | $5.57 \pm 0.02$ |
| Single minimal | $15.02 \pm 0.19$ | $13.87 \pm 0.12$ | $20.52 \pm 0.17$ | $15.13 \pm 0.07$ | $15.48 \pm 0.08$ | $7.81 \pm 0.03$ |
| Single negative | $14.29 \pm 0.19$ | $13.85 \pm 0.16$ | $27.95 \pm 0.33$ | $15.43 \pm 0.17$ | $16.00 \pm 0.13$ | $5.22 \pm 0.02$ |
| Single positive | $16.63 \pm 0.34$ | $13.18 \pm 0.14$ | $27.47 \pm 0.34$ | $14.64 \pm 0.12$ | $16.25 \pm 0.12$ | $5.37 \pm 0.02$ |

Table 4: SSIM and MSE comparison between responsible, random, and nearest neighbor (NN) samples for CelebA dataset using minimal allocations.

random ones. We performed a t-test with the Bonferroni correction. Since we have 16 comparisons we calculated the corrected threshold as $0.05/16 = 0.003125$. Using this correction, all the Responsibility variations showed higher SSIM and lower MSE compared to the random samples, except from the multiple negative activations that did not show a significant difference between two methods. Notably, the nearest neighbor (NN) baseline shows significantly greater similarity compared to both methods. This is expected, as the NN approach selects training samples based on their similarity to the generated images, directly considering MSE. We further note that the values for random and NN differ across activation types. This is because the number of unique responsible samples varies for each row, leading to differences in the corresponding number of random and NN samples.

### 6.4.2 Positive Allocations

Positive allocations point to samples that had the maximum positive influence on weight changes. Similar to the prior setup, we include all four types of activation identifications to identify a subset of weights $W'$, which can be a single or multiple critical weights. We calculate the similarity between our generated images and the responsible samples. Table 5 summarizes the results.

| Activation type | SSIM | | | MSE | | |
|---|---|---|---|---|---|---|
| | **Responsibility** | **Random** | **NN** | **Responsibility** | **Random** | **NN** |
| Multi. maximal | $11.99 \pm 0.18$ | $14.23 \pm 0.15$ | $26.84 \pm 0.32$ | $15.37 \pm 0.11$ | $15.29 \pm 0.13$ | $5.50 \pm 0.02$ |
| Multi. minimal | $12.26 \pm 0.07$ | $14.08 \pm 0.13$ | $18.43 \pm 0.14$ | $15.58 \pm 0.06$ | $15.30 \pm 0.08$ | $9.09 \pm 0.03$ |
| Multi. negative | $12.29 \pm 0.05$ | $14.73 \pm 0.16$ | $27.57 \pm 0.31$ | $13.82 \pm 0.06$ | $15.75 \pm 0.14$ | $5.32 \pm 0.02$ |
| Multi. positive | $11.99 \pm 0.18$ | $14.12 \pm 0.12$ | $26.84 \pm 0.32$ | $15.37 \pm 0.11$ | $15.51 \pm 0.10$ | $5.50 \pm 0.02$ |
| Single maximal | $12.16 \pm 0.05$ | $13.33 \pm 0.10$ | $26.57 \pm 0.31$ | $14.55 \pm 0.06$ | $15.86 \pm 0.09$ | $5.59 \pm 0.02$ |
| Single minimal | $12.33 \pm 0.06$ | $13.97 \pm 0.12$ | $20.74 \pm 0.17$ | $15.56 \pm 0.06$ | $15.50 \pm 0.08$ | $7.71 \pm 0.03$ |
| Single negative | $12.17 \pm 0.06$ | $14.92 \pm 0.20$ | $27.94 \pm 0.33$ | $13.81 \pm 0.06$ | $14.66 \pm 0.06$ | $5.22 \pm 0.02$ |
| Single positive | $11.80 \pm 0.16$ | $14.38 \pm 0.19$ | $27.43 \pm 0.34$ | $15.61 \pm 0.12$ | $15.01 \pm 0.11$ | $5.36 \pm 0.02$ |

Table 5: SSIM and MSE comparison between responsible, random, and nearest neighbor (NN) samples for CelebA dataset using positive allocations.

In contrast to minimal allocations, where responsible samples show a higher similarity to generated images than random, these positive allocations typically show a lower similarity. Using the same corrected p-value threshold, SSIM values were significantly lower for all the variations of Responsibility compared to the random samples. This aligns with our expectations as the output of diffusion models is the opposite of the desired final image; therefore, positive influences on weights should result in less resemblance to the generated images. Although for MSE, while Responsibility variants show higher values in some cases, this is

Generated     Minimal Allocation     Positive Allocation     Nearest Neighbours

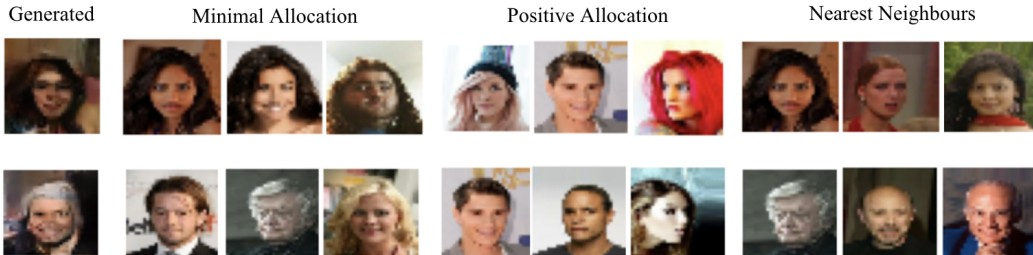

Figure 2: Two generated images using our diffusion model along with their responsible samples derived from minimal and positive allocations along with the baseline nearest neighbor samples.

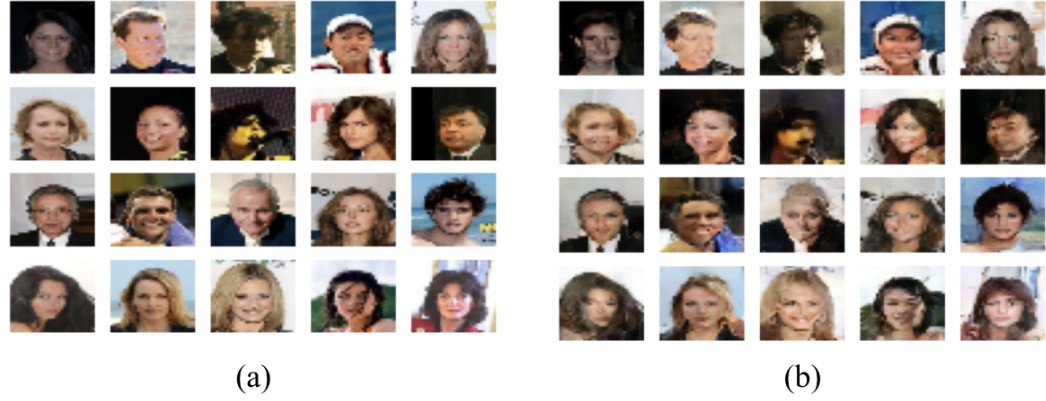

(a)             (b)

Figure 3: (a) Examples of real samples from the test dataset. (b) Reconstructed samples using the diffusion model.

not always the case. The NN baseline again shows the highest similarity, as it is explicitly designed to select the most similar training samples to the generated images.

### 6.4.3   Visual Comparison

In this section, in lieu of a human subject study, we present examples of generated images and their corresponding most responsible samples. Figure 2 shows two handpicked generated images alongside their responsible samples from minimal and positive allocations, as well as the NN samples. The responsible samples derived from minimal allocations demonstrate noticeable similarity to the generated images. In fact, in one case, the responsible sample from the minimal allocation is also the NN sample, demonstrating that the responsible samples can have a strong visual resemblance. The responsible samples derived from positive allocations are not visually similar to the generated images. This is consistent with our quantitative results in Table 5. The NN samples have smaller MSE values as that is how they are selected. However, this similarity might be influenced by secondary factors such as the background rather than core factors such as facial features. In contrast, the responsible samples from minimal allocations show a higher resemblance to the generated images from our perspective.

### 6.5   Reconstruction Experiment Results

In this section, we present the results of an additional experiment where rather than generate new samples using our diffusion model, we reconstruct 20 randomly selected test samples. This allows us to more easily compare to an influence functions (IF) baseline (Koh & Liang, 2017). The reconstructions are shown in Figure 3. Table 6 summarizes the performance of each variant in terms of SSIM and MSE using minimal allocations. Again, we performed a t-test with the Bonferroni correction. Since we have eight tests this time, the p-value threshold is adjusted as $0.05/8 = 0.00625$. The SSIM values that are significantly larger and the MSE values that are significantly smaller are highlighted in bold. We observe that across all Responsibility

types, responsible samples achieve significantly higher similarity to the reconstructed images, as shown by the higher SSIM and lower MSE values for responsible samples compared to samples derived from influence functions. We also conducted the experiment using positive allocations but as expected for diffusion models, Responsibility performed worse compared to influence functions. These results are presented in the appendix.

| Activation type | SSIM | | MSE | |
|---|---|---|---|---|
| | **Resp** | **IF** | **Resp** | **IF** |
| Multiple maximal | $\mathbf{18.73 \pm 0.35}$ | $12.92 \pm 0.13$ | $\mathbf{13.94 \pm 0.09}$ | $18.65 \pm 0.12$ |
| Multiple minimal | $\mathbf{14.23 \pm 0.26}$ | $12.64 \pm 0.14$ | $\mathbf{17.15 \pm 0.10}$ | $18.85 \pm 0.42$ |
| Multiple negative | $\mathbf{15.79 \pm 0.34}$ | $12.75 \pm 0.12$ | $\mathbf{12.30 \pm 0.13}$ | $18.93 \pm 0.10$ |
| Multiple positive | $\mathbf{18.73 \pm 0.35}$ | $12.92 \pm 0.13$ | $\mathbf{13.94 \pm 0.09}$ | $18.65 \pm 0.12$ |
| Single maximal | $\mathbf{17.36 \pm 0.47}$ | $12.74 \pm 0.13$ | $\mathbf{12.35 \pm 0.15}$ | $18.77 \pm 0.11$ |
| Single minimal | $\mathbf{14.29 \pm 0.25}$ | $12.83 \pm 0.34$ | $\mathbf{17.09 \pm 0.09}$ | $18.53 \pm 0.11$ |
| Single negative | $\mathbf{15.36 \pm 0.52}$ | $12.75 \pm 0.10$ | $\mathbf{12.82 \pm 0.18}$ | $19.15 \pm 0.12$ |
| Single positive | $\mathbf{19.00 \pm 0.35}$ | $12.98 \pm 0.15$ | $\mathbf{13.64 \pm 0.13}$ | $18.83 \pm 0.11$ |

Table 6: SSIM and MSE comparison between responsible samples and influence functions samples for CelebA dataset using minimal allocations.

## 7    Conclusions

In this paper, we formally defined the XAI approach Responsibility and evaluated it across several model architectures and domains. The approach improved the performance of a critic model at predicting the behavior of an actor model, was considered by non-expert users to be more understandable than existing XAI approaches, improved user accuracy relative to existing XAI methods and outperformed existing approaches in terms of similarity for image generation tasks. Overall, we view Responsibility as a general XAI approach with the capacity to benefit both humans and machines.

## Acknowledgments

Anonymized

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

# A   Appendix

## A.1   Clarifying Experiments

In this section, we present experiments for improving a reader's understanding of Responsibility. We use MNIST (LeCun et al., 1998) and Fashion MNIST (Xiao et al., 2017) for these experiments due to their simplicity. For most examples, we used LeNet-5 (LeCun et al., 1998) with its 61,706 trainable parameters as the model, as it is simple but reasonably strong. We trained LeNet-5 and tracked the most responsible training values across all layers, though we focus on the first and last layers in these examples. All Responsibility values in this section were computed using the positive allocations and activations.

### A.1.1   Responsibility Values

First, we compute the mean and standard deviation of the changes of the neurons of the final layer, for all training data points. These values show how much each training data point is responsible for the changes of each neuron during the training phase. The mean and standard deviation are 6.84e−10 and 3.36e−4, respectively. The mean is very close to zero while the standard deviation is substantially higher. This follows our intuition that the contribution of a single training data point in a large dataset is likely small. The standard deviation demonstrates that some data points have extreme contribution values. This matches our intuition that we can differentiate a small subset of individual training data points that have a large impact on the final model, which supports the basic concept of Responsibility.

### A.1.2   Inter-class Responsibility

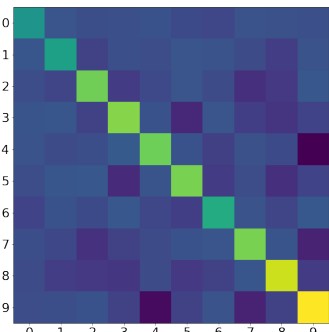

Figure 4: Inter-class Responsibility heatmap. Rows represent classes and columns represent the corresponding neurons.

Next, we inspect the overall impact of particular classes in the training data in terms of the particular neurons used for classification. We expect each training data instance alters all neurons of the last layer during training phase, but each training data instance has a very different impact in terms of the degree of change. We sum up these changes and separate them with respect to the labels of the training instances. We plot this inter-class Responsibility in Figure 4. The rows represent training data classes and the columns represent the neuron associated with a particular class (0-9 for MNIST). We anticipate that training data points of the same class would have the largest impact on the neuron associated with predicting that class, which Figure 4 supports. The maximum Responsibility values for each row and column appear on the diagonal, indicating that training data of a particular class always causes the largest changes in neurons representing the same class. We observe symmetrically low values between the digit pairs (3,5), (2, 7), (4,9), and (7,9). As the two classes in each pair have similar appearances, this likely is due to competition between them. That is, training data points representing the digit "5" decrease the values of the neuron associated with predicting "3", making it less likely for the model to predict "3" when given a 5. Therefore, by visualizing Responsibility as in Figure 4, we obtain a visualization of the model's training process. Specifically, we can view darker and brighter cells as cases where the model was more often incorrect and correct respectively during training.

### A.1.3 Identifying Mislabeled Examples

Here, we investigate the utility of Responsibility for identifying mislabeled training data. We expect mislabeled examples to be among the most responsible data points due to the fact that they should stand out as distinct from other data points of the same class. By the most responsible data points, we mean training samples which have the largest impact on the neurons of the first layer during training. We calculate this impact by considering the absolute changes of the neurons in the first layer. We focus on the first layer over the final layer here as we anticipate the majority of the discriminatory features to be found in the first layer. We use 1000 randomly sampled images from Fashion MNIST to classify T-Shirts and Tops against Shirts. We randomly flip the label for 10% of our training points. We require a simpler model for this example as it involves retraining the neural network a considerable number of times and we also want to ensure that the model would not be robust to mislabeled data. As such, we train a convolutional neural network with one convolutional (8 kernels of size 3x3) and two fully connected layers (sizes of 128 and 2) on this noisy dataset. We then order the 1000 training points both according to their Responsibility as well as randomly to produce 2 sequences. We then iterate through both sequences, representing a manual check of each of the 1000 datapoints. When we reach a particular datapoint, we correct its label if it was incorrect and then retrain the model. Our goal would be to minimize the amount of manual checks required. Figure 5 visualizes this process. The left graph gives the average test accuracy on the Fashion MNIST test set for our model at each index for the Responsibility and Random sequences. The right graph gives the number of "flips" that have been fixed or correct mislabels at each sequence. It is clear that using Responsibility we can identify mislabeled data points sooner than with a random inspection. Thus, we can have a higher test accuracy in early stages of data debugging using Responsibility.

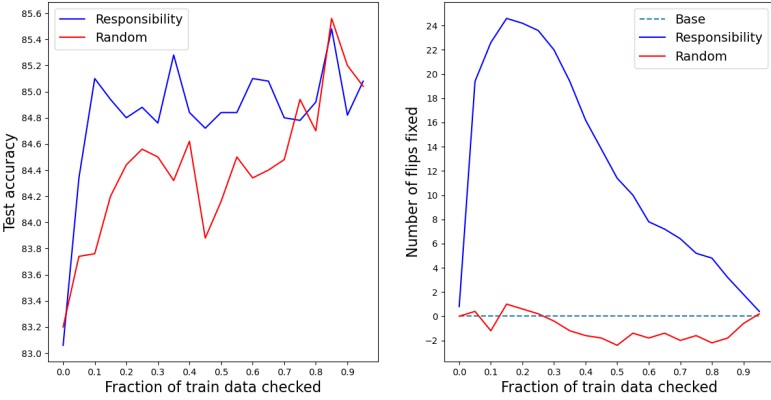

Figure 5: Impact of using Responsibility to inspect and correct mislabeled data in comparison to a random order.

### A.1.4 Understanding Misclassified Examples

Here, we consider how Responsibility can help us better understand classification errors. We again use Fashion MNIST along with the same model as the prior experiment except with the final layer expanded to 10 classes. After training, we used the model to evaluate 600 of the test images, equally sampled across all classes. Even with such a simple model, the total number of misclassified instances is only 70. We visualize four of these 70 examples (left) and their most responsible training instance (right) in Figure 6. Notice that the most responsible instances have similar appearances to the test instance even though they belong to another class. Pairs (C) and (D) are particularly close, with nearly the same color and silhouette. Example (A) in Figure 6 is one of seven pullovers misclassified as a coat. Of these seven, six share the same most responsible training image. A similar pattern can be identified for the other three examples—(B) is one of the seven shirts misclassified as a T-shirt, with five of these sharing the same most responsible image, (C) represents an example of one of the three shirts misclassified as a pullover, with the given most responsible training image being the same for all three and (D) represents one of the seven T-shirts misclassified as

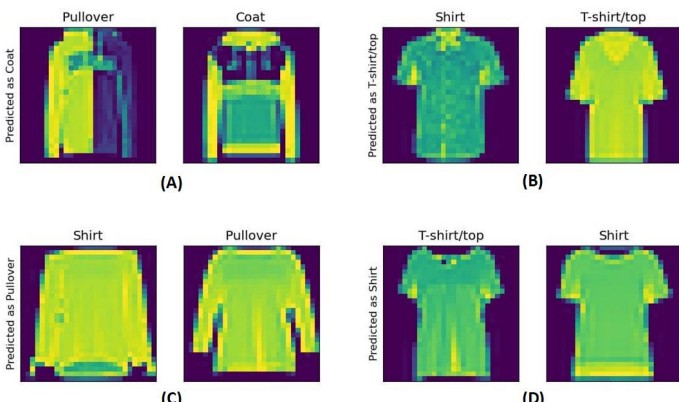

Figure 6: Four randomly selected test samples and their associated most responsible training instance.

a shirt. Its most responsible training instance is shared with five of these seven. This demonstrates how Responsibility can give insight in terms of where our model fails and the kinds of classification problems it struggles with. This could also help in removing training samples that might confuse or negatively impact a model, and may also suggest that these examples could confuse humans as well.

### A.1.5  Responsible Examples Across Different Seeds

One of our assumptions with Responsibility is that different training processes will lead to distinct final models, and that XAI approaches should capture these differences. In this section, we identify Responsibility data points obtained by training the LeNet-5 model on MNIST three times, each using a different random seed. Figure 7 shows these Responsibility data points across the different seeds. For each image, the first rows give the most frequent training data points that make positive changes in the output neurons, the second rows are the training data points which make the largest positive change in the output neurons, the third rows are the most frequent training data points that make negative change in the output neurons and the fourth rows are the training data points which make the largest negative change in the output neurons. Thus, each figure has a total of forty images, corresponding to different ways of measuring Responsibility for the ten final classification neurons of the model across three different random seeds. The first and second rows show that the data points with positive contributions tend to have the same labels as the target labels. However, they also have unusual appearances. This matches prior instance-based results in terms of helping to identify outliers in the training data. The third and fourth rows demonstrate that the data points with the extreme negative contributions do not have the same labels as the target labels but they closely resemble the target labels. While there are shared training instances across all three figures, each also has a unique set. This suggests that Responsibility can capture differences in how each model was trained.

### A.2  Responsibility and Nearest Neighbor Examples

The results of our classification experiments (described in section 4) demonstrated Responsibility could be preferable when training a secondary critic to predict the behavior of a primary actor across two domains, though the user study (described in section 5) complicated this finding. While the NN visualizations harmed the performance of our critics, humans seemed to benefit from them. These visualizations however were not ranked as more understandable than any other method which suggests that NN may have informed users of the correct class, without helping them understand the model's behavior. As an example of this effect, we show two examples in Table 7. In each, we give a test input image from the study ("main") and then the Responsibility image and the NN image. Given that the NN image used a pixel-to-pixel comparison, we found that it typically shared background pixels with the input image. For example, in the top example, the NN image has a blue-white background with ripples as the input image has a blue-white image with curves from vehicle tires. In the bottom example, there are many white pixels in the background of the input image and the NN image. In comparison, the most responsible training instance tends to reflect particular

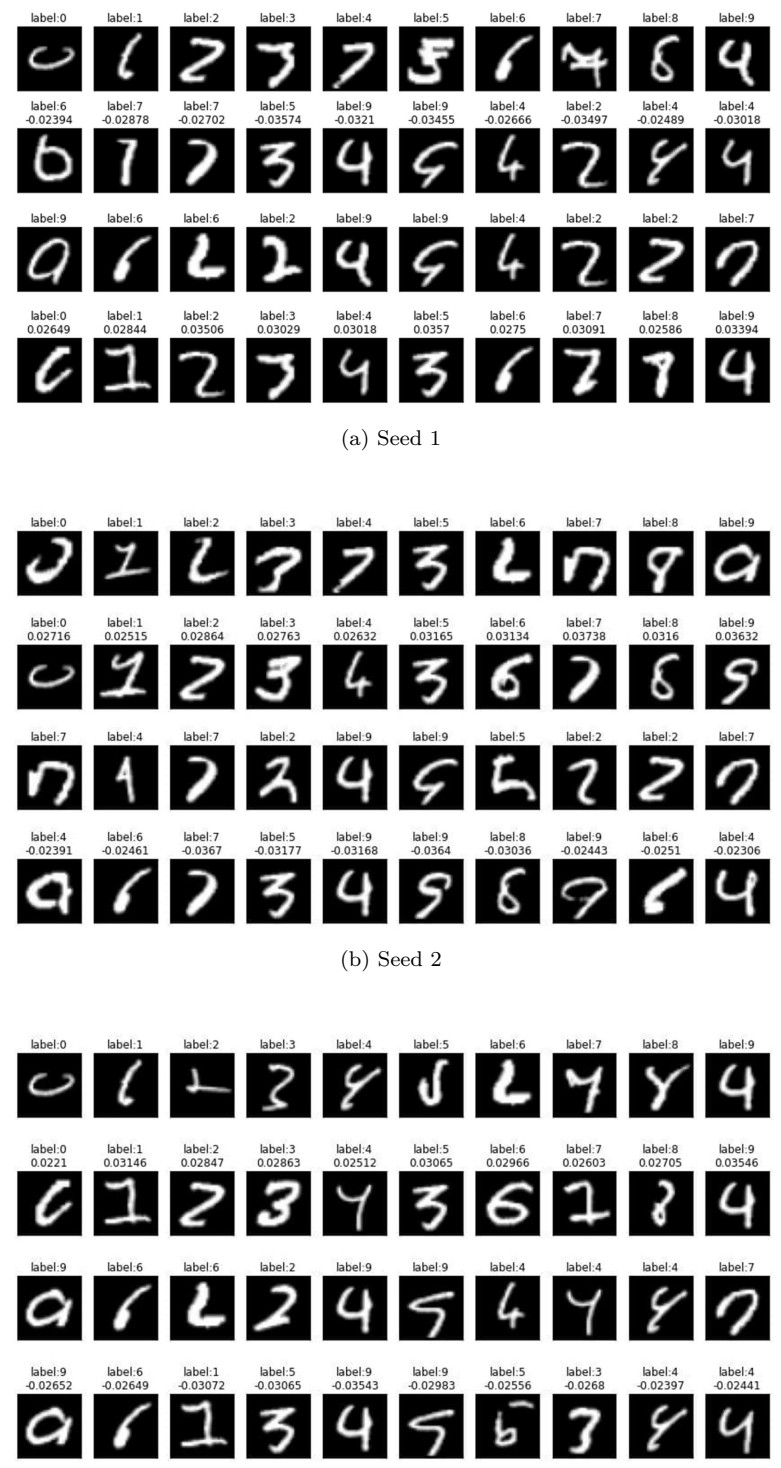

Figure 7: Responsible samples obtained via training on MNIST using three different random seeds.

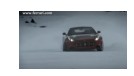 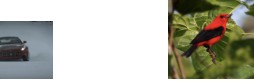 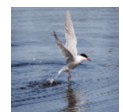

Main Image    Responsibility Image    NN image

(a)

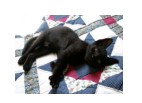 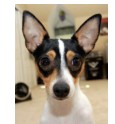 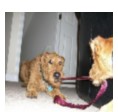

Main Image    Responsibility Image    NN Image

(b)

Table 7: Responsibility images compared to NN images.

features the model used when making a decision. In the top example, the black and red wing matches the front bumper of the car while the black ears and silhouette of the dog match those of the cat. Thus, we view these images as capturing the learned behavior of the model.

### A.3 Oxford Flowers Experiment Results

In this section, we present the results of our Oxford Flowers experiments, focusing on analyzing the performance of our Responsibility variants as in Section 6.4.

### A.3.1 Minimal Allocations

| Activation type | SSIM | | |
|---|---|---|---|
| | **Responsibility** | **Random** | **Nearest Neighbor** |
| Multiple maximal | $2.31 \pm 0.01$ | $2.01 \pm 0.01$ | $6.66 \pm 0.07$ |
| Multiple minimal | $1.82 \pm 0.01$ | $1.91 \pm 0.01$ | $3.05 \pm 0.01$ |
| Multiple negative | $1.73 \pm 0.02$ | $1.60 \pm 0.01$ | $6.66 \pm 0.08$ |
| Multiple positive | $2.31 \pm 0.01$ | $1.89 \pm 0.01$ | $6.66 \pm 0.07$ |
| Single maximal | $2.17 \pm 0.01$ | $2.02 \pm 0.01$ | $6.61 \pm 0.07$ |
| Single minimal | $1.83 \pm 0.01$ | $1.99 \pm 0.01$ | $3.84 \pm 0.03$ |
| Single negative | $2.10 \pm 0.02$ | $1.74 \pm 0.02$ | $7.05 \pm 0.09$ |
| Single positive | $2.45 \pm 0.02$ | $1.65 \pm 0.01$ | $7.00 \pm 0.09$ |

Table 8: SSIM comparison between minimal responsible, random, and NN samples for Oxford Flowers dataset.

| Activation type | MSE | | |
|---|---|---|---|
| | **Responsibility** | **Random** | **Nearest Neighbor** |
| Multiple maximal | $38.99 \pm 0.65$ | $44.54 \pm 0.75$ | $21.88 \pm 0.32$ |
| Multiple minimal | $43.19 \pm 0.67$ | $44.58 \pm 0.62$ | $33.99 \pm 0.46$ |
| Multiple negative | $40.13 \pm 0.58$ | $45.15 \pm 0.78$ | $21.71 \pm 0.31$ |
| Multiple positive | $38.99 \pm 0.65$ | $44.46 \pm 0.83$ | $21.87 \pm 0.32$ |
| Single maximal | $39.09 \pm 0.63$ | $43.95 \pm 0.72$ | $22.01 \pm 0.32$ |
| Single minimal | $43.49 \pm 0.70$ | $44.60 \pm 0.70$ | $30.00 \pm 0.42$ |
| Single negative | $40.31 \pm 0.57$ | $45.18 \pm 0.61$ | $21.13 \pm 0.31$ |
| Single positive | $38.68 \pm 0.67$ | $44.56 \pm 0.72$ | $21.23 \pm 0.30$ |

Table 9: MSE comparison between minimal responsible, random, and NN samples for Oxford Flowers dataset.

Tables 8 and 9 summarize the performance of each variant in terms of SSIM and MSE metrics, respectively. We again ran a Bonferroni corrected t-test and observe that across all activation types except from single and multiple minimal activations, responsible samples achieve significantly higher similarity to the generated

images, as shown by the higher SSIM and lower MSE values for responsible samples compared to random ones. The nearest neighbor baseline again achieves the highest similarity for all variations, as expected.

### A.3.2 Positive Allocations

| Activation type | SSIM | | |
|---|---|---|---|
| | **Responsibility** | **Random** | **Nearest Neighbor** |
| Multiple maximal | $2.03 \pm 0.01$ | $1.92 \pm 0.01$ | $6.65 \pm 0.07$ |
| Multiple minimal | $1.87 \pm 0.01$ | $1.97 \pm 0.01$ | $3.15 \pm 0.01$ |
| Multiple negative | $2.20 \pm 0.01$ | $2.14 \pm 0.01$ | $6.59 \pm 0.07$ |
| Multiple positive | $2.03 \pm 0.01$ | $2.08 \pm 0.01$ | $6.65 \pm 0.07$ |
| Single maximal | $1.90 \pm 0.01$ | $2.10 \pm 0.01$ | $6.63 \pm 0.07$ |
| Single minimal | $1.76 \pm 0.01$ | $1.97 \pm 0.01$ | $3.87 \pm 0.03$ |
| Single negative | $2.15 \pm 0.02$ | $1.93 \pm 0.01$ | $7.05 \pm 0.09$ |
| Single positive | $1.99 \pm 0.02$ | $1.98 \pm 0.02$ | $6.92 \pm 0.09$ |

Table 10: SSIM comparison between positive responsible, random, and NN samples for Oxford Flowers dataset.

| Activation type | MSE | | |
|---|---|---|---|
| | **Responsibility** | **Random** | **Nearest Neighbor** |
| Multiple maximal | $46.37 \pm 0.61$ | $43.35 \pm 0.59$ | $21.95 \pm 0.32$ |
| Multiple minimal | $44.40 \pm 0.70$ | $44.39 \pm 0.66$ | $33.36 \pm 0.46$ |
| Multiple negative | $47.39 \pm 0.77$ | $45.09 \pm 0.79$ | $21.79 \pm 0.31$ |
| Multiple positive | $46.37 \pm 0.61$ | $43.55 \pm 0.66$ | $21.95 \pm 0.32$ |
| Single maximal | $48.02 \pm 0.63$ | $45.39 \pm 0.87$ | $22.08 \pm 0.32$ |
| Single minimal | $44.66 \pm 0.66$ | $44.45 \pm 0.63$ | $29.81 \pm 0.42$ |
| Single negative | $48.34 \pm 0.88$ | $43.79 \pm 0.69$ | $21.16 \pm 0.31$ |
| Single positive | $46.45 \pm 0.66$ | $44.59 \pm 0.77$ | $21.29 \pm 0.30$ |

Table 11: MSE comparison between positive responsible, random, and NN samples for Oxford Flowers dataset.

Tables 10 and 11 summarize the performance of each variant in terms of SSIM and MSE metrics, respectively. A Bonferroni-corrected t-test indicated no significant trend between Responsibility and random samples. In some cases, Responsibility outperforms random samples, while in others, random samples perform better. However, nearest neighbor samples consistently and significantly outperform both Responsibility and random samples as we expected.

### A.3.3 Visual Comparison

We also explore some examples of generated images and their corresponding most responsible samples from the Oxford Flowers dataset. Figure 8 shows two hand selected generated images alongside their responsible samples from minimal and positive allocations, as well as the nearest neighbor samples. As with CelebA, the responsible samples derived from minimal allocations show a resemblance to the generated images, reflecting the influence of these samples on the generation process. In contrast, we cannot see visual similarity between the responsible samples derived from positive allocations and the generated images. The nearest neighbor samples, as expected, are again the most visually similar to the generated images.

|  Generated | Minimal Allocation | Positive Allocation | Nearest Neighbours |
|---|---|---|---|

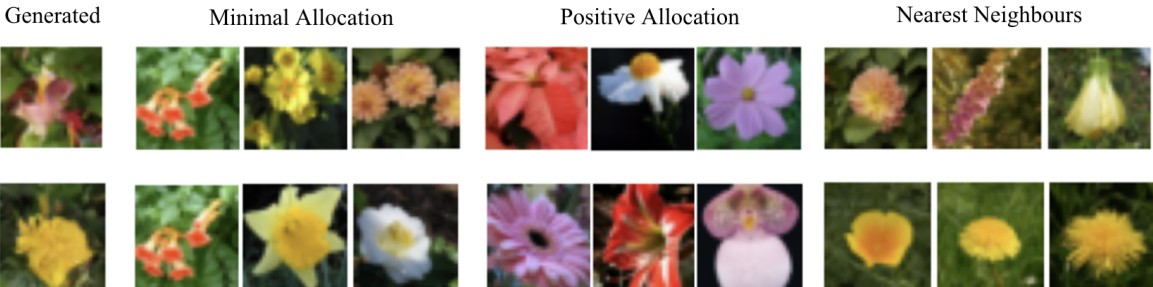

Figure 8: Two generated images using our diffusion model along with their responsible samples derived from minimal and positive allocations. We also include the nearest neighbor samples, which are training samples most similar to the generated image.

## A.4   Maximal and Negative Allocation

Tables 12 and 13 summarize the performance of maximal and negative Responsibility allocations respectively, in terms of SSIM and MSE values. We conducted t-tests with a Bonferroni corrected p-value threshold of 0.003125. The SSIM values that are significantly larger and MSE values that are significantly smaller are highlighted bold in these tables.

| Activation type | SSIM | | MSE | |
|---|---|---|---|---|
| | **Resp** | **Random** | **Resp** | **Random** |
| Multiple maximal | $\mathbf{14.38 \pm 0.17}$ | $13.59 \pm 0.20$ | $16.63 \pm 0.08$ | $\mathbf{16.10 \pm 0.11}$ |
| Multiple minimal | $11.87 \pm 0.06$ | $\mathbf{14.04 \pm 0.13}$ | $15.68 \pm 0.07$ | $\mathbf{15.43 \pm 0.08}$ |
| Multiple negative | $\mathbf{15.07 \pm 0.24}$ | $13.06 \pm 0.14$ | $\mathbf{13.48 \pm 0.16}$ | $15.48 \pm 0.07$ |
| Multiple positive | $\mathbf{14.38 \pm 0.17}$ | $13.27 \pm 0.16$ | $16.63 \pm 0.08$ | $\mathbf{16.10 \pm 0.14}$ |
| Single maximal | $\mathbf{14.91 \pm 0.20}$ | $14.36 \pm 0.17$ | $\mathbf{15.02 \pm 0.09}$ | $15.49 \pm 0.09$ |
| Single minimal | $11.79 \pm 0.06$ | $\mathbf{14.04 \pm 0.12}$ | $15.73 \pm 0.06$ | $\mathbf{15.43 \pm 0.07}$ |
| Single negative | $\mathbf{15.02 \pm 0.24}$ | $13.71 \pm 0.17$ | $\mathbf{13.47 \pm 0.16}$ | $16.26 \pm 0.15$ |
| Single positive | $14.33 \pm 0.18$ | $14.37 \pm 0.19$ | $16.89 \pm 0.08$ | $\mathbf{16.05 \pm 0.13}$ |

Table 12: SSIM and MSE comparison between maximal allocation Responsibility samples and random samples.

| Activation type | SSIM | | MSE | |
|---|---|---|---|---|
| | **Resp** | **Random** | **Resp** | **Random** |
| Multiple maximal | $13.67 \pm 0.17$ | $\mathbf{13.92 \pm 0.20}$ | $15.26 \pm 0.08$ | $15.25 \pm 0.09$ |
| Multiple minimal | $13.59 \pm 0.12$ | $\mathbf{14.05 \pm 0.13}$ | $\mathbf{14.91 \pm 0.08}$ | $15.38 \pm 0.06$ |
| Multiple negative | $\mathbf{15.13 \pm 0.21}$ | $12.66 \pm 0.15$ | $\mathbf{12.73 \pm 0.15}$ | $16.44 \pm 0.12$ |
| Multiple positive | $\mathbf{13.67 \pm 0.17}$ | $13.12 \pm 0.08$ | $\mathbf{15.26 \pm 0.08}$ | $16.24 \pm 0.09$ |
| Single maximal | $14.51 \pm 0.19$ | $14.34 \pm 0.11$ | $\mathbf{13.96 \pm 0.10}$ | $15.02 \pm 0.07$ |
| Single minimal | $13.50 \pm 0.12$ | $\mathbf{13.97 \pm 0.13}$ | $\mathbf{15.12 \pm 0.08}$ | $15.52 \pm 0.08$ |
| Single negative | $\mathbf{15.18 \pm 0.23}$ | $13.18 \pm 0.08$ | $\mathbf{12.65 \pm 0.15}$ | $15.12 \pm 0.07$ |
| Single positive | $13.65 \pm 0.17$ | $13.69 \pm 0.19$ | $15.42 \pm 0.09$ | $15.20 \pm 0.06$ |

Table 13: SSIM and MSE comparison between negative allocation Responsibility samples and random samples.

We observe that across these Responsibility allocation types, there is no consistent trend in the similarity between responsible samples and the generated images. In some cases, responsible samples achieve higher

similarity to the generated images, as indicated by higher SSIM and lower MSE values. However, in other cases, random samples outperform responsible samples in similarity metrics. These inconsistent results suggest that these Responsibility variants may select training samples similarly to random selection, rather than identifying truly influential examples.

| Activation type | SSIM | | MSE | |
|---|---|---|---|---|
| | **Resp** | **IF** | **Resp** | **IF** |
| Multiple maximal | $9.80 \pm 0.19$ | $\mathbf{12.83 \pm 0.14}$ | $19.46 \pm 0.18$ | $\mathbf{18.67 \pm 0.12}$ |
| Multiple minimal | $11.18 \pm 0.11$ | $\mathbf{12.73 \pm 0.15}$ | $\mathbf{17.50 \pm 0.09}$ | $18.69 \pm 0.10$ |
| Multiple negative | $9.78 \pm 0.09$ | $\mathbf{12.79 \pm 0.12}$ | $\mathbf{14.36 \pm 0.05}$ | $18.92 \pm 0.11$ |
| Multiple positive | $9.80 \pm 0.19$ | $\mathbf{12.83 \pm 0.14}$ | $19.46 \pm 0.18$ | $\mathbf{18.67 \pm 0.12}$ |
| Single maximal | $10.10 \pm 0.08$ | $\mathbf{12.76 \pm 0.12}$ | $\mathbf{16.36 \pm 0.15}$ | $18.64 \pm 0.10$ |
| Single minimal | $11.38 \pm 0.12$ | $\mathbf{12.54 \pm 0.51}$ | $\mathbf{17.29 \pm 0.06}$ | $18.35 \pm 0.11$ |
| Single negative | $9.97 \pm 0.12$ | $\mathbf{12.73 \pm 0.10}$ | $\mathbf{13.99 \pm 0.07}$ | $19.10 \pm 0.12$ |
| Single positive | $10.54 \pm 0.22$ | $\mathbf{12.98 \pm 0.15}$ | $18.98 \pm 0.23$ | $18.87 \pm 0.11$ |

Table 14: SSIM and MSE comparison between responsible samples and influence functions samples for CelebA dataset using positive allocations.

### A.5 Diffusion Model Reconstruction Experiment using Positive Allocations

Here, we present results for the reconstruction experiment (described in section 6.5) using positive allocations instead of minimal allocations. Table 14 summarizes the results of each variant in terms of SSIM and MSE metrics. We again performed a t-test with the bonferroni corrected p-value threshold of 0.00625. Significantly different values are highlighted in bold. As expected for diffusion models, we observe that positive allocation Responsibility variants result in lower SSIM values for the reconstructed images compared to samples derived from influence functions. However, the trends in MSE values are less consistent: in some cases, Responsibility variants exhibit lower MSE, while in others, they do not. This variability suggests that Responsibility may identify samples with pixel values that closely match the reconstructed images (leading to lower MSE) but lack the structural coherence or features necessary for high perceptual similarity, as measured by SSIM.

### A.6 Explaining a Reinforcement Learning Agent

To further demonstrate the generality of Responsibility as an XAI approach, we include an example of a Reinforcement Learning (RL) agent trained in the Morai Maker environment (Guzdial et al., 2019). We include a visualization of the Morai Maker interface in Figure 9. It is a video game level editing tool, in which a human user and an AI agent take turns making changes to a level based on the classic video game *Super Mario Bros.* It was created for the purpose of testing human-computer interaction for video game level editing. Two human subject studies have been run on the tool, which we refer to as "Study 1" and "Study 2". More detail on both studies, the tool itself, and the backend AI agent can be found in (Guzdial et al., 2019). For this work, we use data from the two studies to approximate the impact of applying Responsibility to the backend RL agent.

We use Responsibility for tracking the training instances that maximally alter each neuron of the Deep Q network described in (Guzdial et al., 2018). The difference is that rather than an image, a training instance here represents one set of state, action, and reward values. A state here is a *Super Mario Bros.* level or an in-progress level. We can associate each state, action, and reward value with the sequence of states, actions, and rewards that represent the level design process, going from an initially blank level to a complete level. Since the first convolutional layer is the layer that most directly reasons over the level structure, we make use of most responsible training instances from this layer for these experiments. We give an example of the most responsible state for a particular test state and agent action in Figure 10. On the left we have the the current state and the AI agent's action, adding the entities on the lower left to the top left state. On the

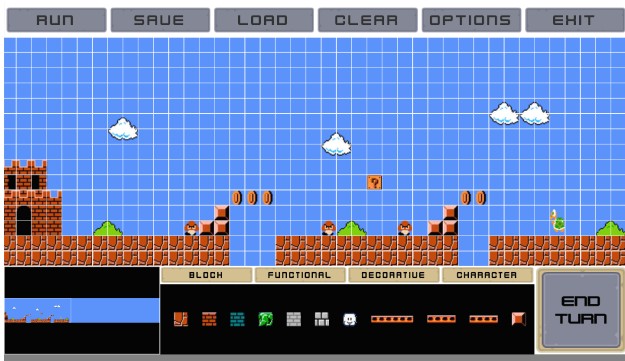

Figure 9: The Morai Maker interface, reproduced with permission from (Guzdial et al., 2019).

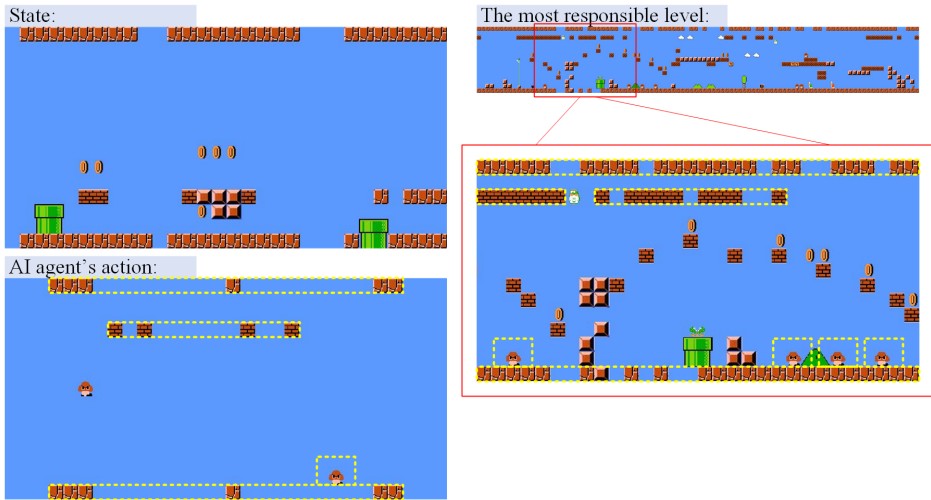

Figure 10: Explaining an AI agent's action by representing the most responsible state from the agent's training history. Similar regions are highlighted.

right we give a final level that the RL agent constructed with a human partner during training and the state portion of the most responsible training instance. We consider the final level associated with a particular most responsible state to be the "most responsible level".

We require a baseline for comparison purposes in these experiments. Due to the lack of general, easily applied RL XAI approaches, we make use of a baseline of 20 randomly selected states from the training history of the RL agent. Thus, for each of the following experiments, when we employ our Responsibility approach we identify the one most responsible training instance in comparison to these 20 randomly selected instances. For computing Responsibility values, we use positive allocations and activations. This setup is designed to put our approach at a disadvantage and make up for the weakness of a typical random baseline, since we are comparing only one Responsibility instance to 20 random instances.

For comparison purposes, we require a metric that can compare two states- two levels for similarity. We have the "Study 1" and "Study 2" data but did not run an extra human subject study due to lack of resources. Thus we instead make use of a quantitative measure of similarity meant to approximate certain conditions we might want to see in an explanation for two different use case. We desired a metric that detects local overlaps between states. We wanted to base the metric on comparing sliding windows of each state, windows which are not the same size as the first convolutional layer of our RL model, to capture some local structure without biasing the metric too far towards our first convolutional layer. As a result, we found all 3x3 non-empty patches for both input states. Then we counted the number of exact matches of these patches on both sides, removing the matched ones from the pool since we wanted to count the same patches only once. Finally, we divided the total number of the matched patches by the total number of unique patches in the states. We refer to this metric as the *local overlap ratio*. A ratio of 0 represents no overlap between pairs of states while a 1 represents an exact match.

We used this metric and the above baseline in two experiments—1) a "prediction experiment" to evaluate the ability of Responsibility to help users predict the behavior of the model and 2) a "user labeling error experiment" to evaluate the ability of Responsibility to help users avoid misunderstanding the model.

### A.6.1 Prediction Experiment

In both "Study 1" and "Study 2", the AI agent partner works with a number of different humans to design *Super Mario Bros.* levels. In this first experiment we study if Responsibility can be used to help users predict the behavior of the AI partner. For this, we used test sets from "Study 1" and "Study 2". Each test set is made up of a sequence of alternating human and AI actions, where each action is making changes to an in-progress level. On each action, we examine the effect of presenting the most responsible level, the final level associated with a most responsible training instance, to the human partner. In particular, we want to determine how similar this most responsible level is to the level after the AI agent's next action. If they are more similar, this would indicate that the most responsible level would help a user better predict the AI agent's actions. We used the local overlap ratio and the random baseline described above.

We had 242 samples in the "Study 1" test set and 69 samples in the "Study 2" test set since "Study 1" had roughly five times as many participants. Since we wanted to compare instances in which the AI agent actually made some significant changes, we chose instances where the AI agent added more than 10 entities in its next action. Thus we ended up with 38 and 46 instances from the first and second test sets, respectively.

Our approach outperforms the random baseline for 78.94% of the "Study 1" test instances and for 67.29% of the "Study 2" test instances. The average of the local overlap ratios is shown in Table 15 (higher is better). The minimum value here would be 0 for zero overlap and the maximum value would be 1 for complete overlap between the AI agent's next action and the most responsible level or the random levels. This normalization means that even small differences in this metric represent large perceptual differences. For example, a 0.04 difference in the local overlap ratio between the most responsible level and the random levels indicates the most responsible level has 20 more 3x3 non-empty overlaps. We expect that the reason that the "Study 2" values are generally lower is that the second study had published level designers as participants rather than novices, meaning that there was more varied behavior compared with "Study 1".

| Test Set | Most Responsible Level | Random Levels |
|----------|:----------------------:|:-------------:|
| Study 1  | **0.4653**             | 0.3841        |
| Study 2  | **0.2880**             | 0.2472        |

Table 15: Average local overlap ratio of the most responsible levels compared to random levels for both test sets.

### A.6.2 User Labeling Error Experiment

For the second experiment, we focused on whether Responsibility could assist a user in better understanding good and bad agent actions during the human-AI creation process. Specifically, during this process, a human user might delete an addition made by the AI only to later add the same content back themselves. Alternatively, a user could keep an addition by the AI at first, only to delete it later. We refer to these instances as *False-positive* and *False-negative* decisions from the human user, respectively. If we could help the user avoid making these kinds of decisions, it may streamline the level creation process. We anticipated that one reason that users made these kinds of decisions was from a lack of context for the AI agent's action. Thus, if the user had an explanation they may not make these decisions.

We implemented an algorithmic way to determine false-positives and false-negatives among the two test sets described in the previous experiment. We first find all user decisions in terms of deleting or keeping an addition by the AI agent. Then we look at the level at the end of the user and the AI agent's interaction. If a deleted AI addition exists in the final level, it is counted as a false-negative (FN) example, and if a kept addition does not exist in the final level, it is counted as a false-positive (FP) example. Once we discovered all FN and FP examples, we found the state before associated content was added by the AI agent and named it the *Introduction-state (I-state)*. We found the state in which false-positivity or false-negativity occurred (i.e. when a user re-added a FN or deleted a FP) and named it the *Contradiction-state (C-state)*. Since some change between the I-state and the C-state led to the user altering their decision, we wanted to see some sign that presenting the most responsible level to the user could change their mind before they reached the C-state. Thus, we compared these two states to find all the changes that the AI agent and human user made and named this the *Difference-state (D-state)*. We compared each D-state with the final generated level derived from the most responsible training instance. We also compared each D-state with 20 other randomly selected levels from the existing data as our random baseline. For the comparison, we used the local overlap ratio as above. If Responsibility is more similar to the D-state than the Random baseline, we will be able to say that there is some support that the most responsible level might help the user avoid FPs and FNs in comparison to random levels.

We found 5 FN and 24 FP examples for the "Study 1" data and 5 FN and 54 FP examples for the "Study 2" data. We anticipate that "Study 2" had more FPs as the users were published level designers and so were more likely to have their own vision for the final level, making it more likely that they would eventually delete the additions made by the AI, even if they kept them at first. For the first test set, our approach outperformed the random baseline in 65.51% of the examples. The average of the local overlap ratio values for our approach was 0.1717 which is more than the 0.1647 for the random levels. For the second test test, our approach outperformed the baseline in 59.32% of the examples. The average of the local overlap ratio values were 0.2665 and 0.2328 for the most responsible level and random levels, respectively. Again this represents a large perceptual difference of roughly 15 more non-empty 3x3 overlaps.

Interestingly, our approach outperforms the random levels in all of the FN examples in the second test set, compared with just 20% of FN in the first test set. Further, our approach performs around 1.5 times better than the random levels in 15 of the FP examples in the second test set. This may suggest that the Responsibility information would be especially helpful in terms of helping level design experts make decisions more quickly. However, we note this is all approximation, and while this is a positive trend in terms of the applicability of Responsibility to human-AI level design co-creation, a human subject study would be required to confirm these results.

