# OpenReview forum: "Responsibility: A General Instance and Training Process-based Explainable AI Approach"
_TMLR — Rejected by TMLR_

### Review · Reviewer_Lo6m · 2025-07-23

**Summary Of Contributions:**

This work aims at understanding and explaining models from a training data perspective. Specifically, it proposes the method Responsibility that estimates the contribution of individual training samples to the final model by tracking their influence on parameter updates in training. It approximates each sample’s effect on a weight update by allocating the total update in proportion to the sample’s loss within a training batch. At inference, the method identifies the most important weights, then samples contributing the most update at those weights are considered influential. This method is agnostic to model architecture, and the paper provides experiments on both classification and generative settings.

**Audience:**

Yes

**Broader Impact Concerns:**

No broader impact concerns

**Claims And Evidence:**

No

**Requested Changes:**

1. Further justification for estimating per-sample contribution by proportion in $L_{\text{batch}}$:

The total weight update $w_i'-w_i$ results from aggregating over gradients from all $b$ samples in the batch, i.e., $\nabla_wL_1+...+\nabla_wL_b$, hence change in weight $w_i$ due to sample $j$ is $\Delta w_{i,j}=\nabla_wL_j$. In literature, TracIn and its extension to generative models [1, 2] explicitly models parameter update as $w_{k+1}-w_{k}=-\eta_t\nabla L(w_k,z_k)$. While the proposed method directly uses $\Delta w_{i,j} = \frac{L_j}{L_{\text{batch}}}(w_i'-w_i)$ , which seems like a loose approximation. I believe the method would strengthen by further explanation or (a). empirical evidence showing correlation between this loss-based attribution and actual per-sample gradients, (b). comparison against a natural baseline using $\nabla_wL_j$

2. Additional baselines: Paper only compares to limited other methods, and NN doesn't seem like a strong baseline given that Actor is not of high accuracy. This work will be strengthened if more recent baselines are compared to it, as current improvement is not as clear.

3. I am also curious if there are any patterns in the selected critical weights from 3.2 Activation Identification (e.g. Are some weights frequently selected, or does it tend to be diverse? What are the weights?)


[1] Pruthi, G., Liu, F., Kale, S., & Sundararajan, M. (2020). Estimating training data influence by tracing gradient descent. Advances in Neural Information Processing Systems, 33, 19920-19930.

[2] Xie, T., Li, H., Bai, A., & Hsieh, C. J. (2024). Data attribution for diffusion models: Timestep-induced bias in influence estimation. arXiv preprint arXiv:2401.09031.

**Strengths And Weaknesses:**

Strengths:

1. Proposed method is simple and does not depend on any specific model architecture, broadly applicable across various domains.
2. Covers a wide range of experiment settings, including classification, generative, and human subject study.
3. Evaluation using a predictor model (critic) is interesting and aligns with the XAI objective.
4. Clearly written

Weaknesses:

1. Heuristic for the proposed method seems weak; current theoretical or empirical justification is limited
2. Method has many variants in Allocation and Activation type, which may require particular domain knowledge and difficult for off-the-shelf usage
3. Lack of comparison with recent works in influence estimation / data attribution
4. Performance gains over other methods appear modest. For example, Figure 1 seems to show Influence Functions may provide more intuitive explanations when used as a stand-alone interpretation.

---

> ### Author Response · Authors · 2025-08-08
>
> We thank you for the insightful comments and feedback which we address below:
>  - We wish to clarify that the theoretical basis for our work is inspired by prior work in transfer learning in terms of choice of layer choice and batch-based optimization in terms of proportional impacts of the loss. We will clarify this theoretical basis in the revised version of the paper.
>  - We acknowledge that we propose many different allocations and activations. However, we feel that our empirical results present clear use cases for classification (positive allocations being most useful) compared to diffusion (negative allocations) and that we can extend this as a decision making mechanism for new applications. We will clarify this in the revised version of the paper.
>  - Thank you for your suggestions of prior work to compare against. We apologize for not being familiar with these and will consider adding comparisons against them in the revised version of the paper, in addition to adding these to the related work.
>  - While we agree that the paper would be strengthened by comparisons against the suggested baselines, we argue that nearest neighbors is an appropriate baseline given its strong performance in several of the experiments, particularly in human-facing evaluations.

---

### Review · Reviewer_gvY4 · 2025-07-24

**Summary Of Contributions:**

The paper named "Responsibility: A General Instance and Training Process-based Explainable AI Approach" introduces Responsibility, a process‑aware, instance‑level attribution method that quantifies how much each training example contributes to the final prediction by tracking weight updates throughout training, selecting inference‑critical weights (via positive/negative/maximal/minimal activation rules), and mapping those weights back to the most “responsible” training instances.

The approach is shown to be architecture and task‑agnostic. Experiments performed on image classification and generation tasks show good performances. Moreover, were performed experiments involving an user study focused on non-experts users.
FInally, more evidence and experiments about the presented method are presented in the appendix

**Audience:**

Yes

**Broader Impact Concerns:**

Although the paper includes a user study, I do not think the ethical implications rise to the levle that would require an additional Broader Impact Statement. The protocol appears minimal‑risk and does not handle sensitive personal data.

**Claims And Evidence:**

Yes

**Requested Changes:**

- The paper should provide explicit computational and memory complexity.

- Ablations with realistic batch sizes greater than one, common optimisers like Adam or AdamW, and typical schedulers are needed to verify that allocation remains stable and useful

- Although the method targets a single training run, the paper does not assess variability across seeds. Including a short auxiliary experiment in the appendix, e.g., comparing the selected instances across runs, would help clarify the method’s stability.

- The selection of W′ should be formalised and empirically tested and the sensitivity of results to this choice should be reported.

- Comparisons with state-of-the-art data attribution methods such as TRAK and Datamodels should be added to at least one classification setup, or at least better explain why these methods were not selected.

- Additional visual schematics or graphics would help users better understand the approach and its underlying structure. A such as visual and practical explainations should guide practitioners in choosing among positive, negative, maximal, and minimal allocation or activation variants per domain

**Strengths And Weaknesses:**

Strenghts:
- The work is conceptually clear and with a limited novelty because it shifts the focus from post‑hoc analysis of the final model to a training‑process‑aware perspective: this aspect has already been explored but they present an interested method
- The formalism is well‑modularised, with three clean stages, allocation, activation identification, and mapping, that enable modular variants and domain‑specific choices.
- The evaluation is multi‑perspective, combining a critic‑vs‑actor setup with standard metrics on CIFAR‑10 and PL04, a large-scale user study involving 300 participants supported by appropriate statistical tests (e.g. MANOVA), and diffusion-model analyses using SSIM and MSE with multiple-comparison control (e.g. Bonferroni), alongside comparisons to neural networks and implicit functions.
- The paper clearly presents its limitations by explicitly highlighting scenarios where the Responsibility approach does not outperform alternatives, such as in pixel‑wise similarity where neural networks trivially dominate, and in several diffusion model variants under random conditions.
- It demonstrates additional use cases that highlight the actionability of the approach, including applications in data debugging (e.g., identifying noisy labels), understanding model errors, and a reinforcement learning scenario, all of which are detailed in the appendix.

Weakness:
- The paper under-analyzes scalability in terms of time and memory, as storing per-weight, per-example responsibility becomes quickly infeasible for modern-scale models such as large language models, vision transformers, or high-resolution diffusion models, and no rigorous complexity analysis or compression strategy is provided.

- The batch size is set to 1 in the PL04 and CIFAR-10 experiments, while the proportional-to-loss rule proposed in Equation 3 for handling larger batches is not empirically stress-tested, leaving uncertainty about its behavior with standard optimizers (e.g. Adam, AdamW), common schedulers, and realistic batch sizes. This is a problems treated in similar papers such as TracIn or TRAK.

- The selection of the “critical” layer or subset′ W′ appears heuristic, with no principled protocol for choosing it, such as using percentiles or statistical thresholds, and no sensitivity analysis is provided to assess robustness across different layers. I understand that this approach is also used in papers such as TracIn, but a more deep analysis will strenght the methodology.

- The paper lacks direct comparisons with recent data-attribution methods, such as TRAK, Datamodels, which limits the contextualization of its contributions within the current state of the art.

---

> ### Author Response · Authors · 2025-08-08
>
> We thank you for the insightful feedback and address your comments below:
>  - We acknowledge lack of analysis of memory requirements for storing the responsibility arrays. This is dependent on the total number of weights that we consider |W| and the total number of training samples |T|, thus requiring a table |W|*|T| in size. We will clarify this in the final paper. The reviewer is correct that this becomes untenable if storing values for all weights for large transformer models. However, as we point out in the paper, we do not always need to track changes across all weights. We take inspiration from transfer learning work and focus only on the final layers for classification tasks, for example. We believe similar approaches could allow for application to larger models. We will add this discussion of the computational and memory requirements to the section of the paper discussing the Responsibility approach.
>  - We acknowledge that we have not empirically stress-tested the proportional-to-loss rule. However, we point out that using a batch size of 1 essentially turns other optimizers into stochastic gradient descent. As such, we feel that the proportional-to-loss rule is at least theoretically justified considering the differences in gradient estimation between SGD and other optimizers like Adam. We will clarify this argument in the revised version of the paper.
>  - We chose the last layer as the critical layer following the typical approach in finetuning and transfer learning applications. We will clarify this in the paper.
>  - We appreciate the suggestion of TRAK and Datamodels. We were not familiar with these approaches but will consider them for comparison in the revised version of the paper.

---

### Review · Reviewer_VbGR · 2025-07-25

**Summary Of Contributions:**

This paper presents "Responsibility," an XAI approach that identifies the most responsible training instance for a model's decision by tracking weight changes during training. The method consists of three steps: (1) responsibility allocation using proportional loss-based attribution of weight changes to training samples, (2) activation identification to find critical weights during inference, and (3) responsibility-activation mapping to connect training instances to model decisions. The authors evaluate the approach across multiple domains (image classification, text classification, diffusion models, and reinforcement learning) using both automated metrics and a 300-participant Amazon Mechanical Turk study. The key claim is that this training-process-based approach provides more actionable explanations than existing post-hoc methods by answering "what did the AI learn that led to this decision?"

**Audience:**

Yes

**Broader Impact Concerns:**

could be added given that XAI methods can significantly influence how AI systems are deployed and trusted:
- false confidence in explanations, especially if limitations unclear
- could amplify biases - if training process itself embeds biases

**Claims And Evidence:**

No

**Requested Changes:**

## Critical for acceptance:

theory
- provide theoretical justification for the loss-proportional allocation scheme. Compare against alternative allocation methods (gradient-based, uniform, etc.) with both theoretical analysis and empirical validation.

experiments:
- use realistically accurate models rather than intentionally degraded ones
- implement proper influence function baselines or justify why approximations are necessary
- include additional training-process-based baselines from recent literature

- provide more nuanced interpretation of mixed results in the human study
    - separate prediction accuracy from understanding measures

- address scalability concerns: analyze computational and memory requirements, maybe demonstrate on larger more realistic model/dataset, and add guidelines for layer selection

## Would strengthen the work:
- improve diffusion model evaluation by developing more appropriate metrics for explanation quality in generative settings, potentially through additional human evaluation.
- add comprehensive ablation studies examining different allocation strategies, activation identification methods, and their interactions.
- provide clearer practical guidelines for applying the method across different domains and architectures.
- better position the work relative to other training-process-based interpretability methods.

**Strengths And Weaknesses:**

## Strengths:

- Novel perspective on training-process explanations: The focus on tracking actual weight changes during training rather than approximating influence post-hoc is conceptually appealing and offers a different lens on instance-based explanations.
- The authors demonstrate the method across diverse domains (classification, generation, RL) and architectures, showing broader applicability than many XAI methods.
- The 300-participant AMT study provides valuable empirical evidence about user preferences and understanding, which is often missing in XAI papers.
- The responsibility allocation mechanism using proportional loss attribution (Equations 1-5) is well-defined and implementable.


## Weaknesses:

Theory:
The core assumption that loss-proportional allocation (Equation 3) accurately reflects training instance influence lacks theoretical grounding. Why should $\frac{L_j}{L_{batch}}$ be the correct weighting? Alternative allocation schemes (gradient magnitude, second-order information) are not explored or justified against.

Experiments:
- Intentionally training low-accuracy models (60% on CIFAR-10) makes claims about real-world applicability questionable
- The critic-actor evaluation paradigm may be a weak proxy for human understanding
- Batch size of 1 for "simplification" eliminates the core challenge the method aims to solve
- The baselines appear inadequate:
	- Influence functions implementation appears suboptimal (using Charpiat et al. approximation rather than addressing computational issues)
	- Missing comparisons to other training-process-based methods
	- Nearest neighbor baseline is trivially weak
- In the human user study:
    - "Understanding" is conflated with prediction accuracy
    - Statistical analysis lacks important controls (order effects, task difficulty)
    - Results show mixed evidence: Responsibility ranks 2nd in preference but users achieve highest accuracy with nearest neighbor
- For the diffusion model evaluation, using image similarity metrics (SSIM, MSE) as proxies for explanation quality is problematic since these don't measure actual explanatory value for generative models.

- No analysis of memory requirements for storing responsibility arrays or computational overhead during training.
- The choice of which layer's weights to analyze appears ad-hoc and domain-dependent without principled guidelines.
- No systematic investigation of different allocation strategies or activation identification methods.

Writing:
- some overclaim:
	- "outperforms existing approaches" → "shows competitive performance with existing methods in our experimental settings"
	- or "improves human understanding" → "shows promise for improving user prediction accuracy in certain contexts"
- maybe adding a dedicated limitations and assumptions section (3.4) would help:
	- loss-proportional allocation as a heuristic approximation (that may not capture all aspects of training influence)
	- method requires storing additional data structures during training, increasing memory, address this overhead: O(n × w) where n=training samples, w=weights
	- layer selection is currently domain-dependent and requires empirical validation
	- performance may vary with batch size (acknowledge that batch size=1 is unrealistic)
	- "While we acknowledge that 60% accuracy on CIFAR-10 is below practical deployment thresholds, this limitation allows us to create a balanced dataset for the critic evaluation. Future work should validate these findings on higher-performing models."
	- Clearly state upfront that the critic-actor paradigm is a proxy measure with known limitations
- related work may be easier to follow with an explicit table comparing the following: 1. training-process methods: Responsibility (this work), influence functions, gradient-based tracking, 2. post-hoc methods: LIME, SHAP, GradCAM with the key differences when information is captured, computational requirements, types of insights provided
- in the discussion you could position it a but more nuanced: when does responsibility work well, when not, and position more as complementary to XAI toolkit rather than replacement
- in introduction, lead with specific contribution (training-process tracking) rather than broad XAI claims
	- make it less overclaim: "exploring the potential of" rather than "demonstrating superiority of" and state scope limitations upfront
- write a bit more precisely: for example, replace "significantly better" with specific metrics and confidence intervals
- and use appropriate scientific hedging: "suggests" or "indicates" rather than "demonstrates" for weak evidence

## Minor:
- tables look better without the column rules, best use booktabs with toprule, midrule and bottomrule
- table captions should be above tables (TMLR standard), currently inconsistent
- add explanatory text to equation 3 (for justification): something like, "We use loss-proportional allocation as a first-order approximation of training influence, based on the intuition that samples with higher loss contribute more to weight updates. While this may not capture all aspects of influence (such as gradient direction or higher-order effects), it provides a computationally tractable approach that aligns with standard gradient descent dynamics."
- use terms consistently: "Responsibility" vs "responsibility", "Training instance" vs "training sample" vs "training example", "Weight change" vs "weight update", "XAI approach" vs "XAI method", some citations use full author names, others abbreviated (e.g., "Koh & Liang" vs "Khadivpour and Guzdial")
- references lists is sometimes incomplete (years missing), or inconsistent

---

> ### Author Response · Authors · 2025-08-08
>
> Thank you for your insightful feedback and comments. We address these below:
>  - Regarding training low accuracy models on CIFAR-10, we want to clarify that the goal of this experiment was to demonstrate the utility of the approach for human users rather than its real-world applicability.
>  - While one could argue that the critic-actor evaluation is not a strong proxy for human understanding, we argue that the results of the human subject study suggest that it is indeed a reasonable proxy.
>  - Regarding the comment on baselines, the Charpiat et al. approximation is standard practice for comparing with influence functions. Additionally, while the nearest neighbor baseline might be weak in terms of technical implementation, it is an informative baseline for human-facing tasks, as demonstrated in the results. Finally, we would appreciate it if you could recommend any additional training process-based methods that would be appropriate to compare our approach against.
>  - Referring to the human user study, our intention was not to conflate 'understanding' with prediction accuracy but to use the latter as a signal for understanding. We apologize for the confusion and will improve the related text in the paper to make this clearer.
>  - In terms of the statistical analysis lacking important controls, we wish to first clarify that the user study is balanced across the different orderings, and that a MANOVA showed no significant impact of ordering effects on any of the measured variables. We do assume equivalent task difficulty, but we believe this to be a reasonable assumption given the small number of classes and the lack of ambiguity in the images (see examples in the paper). We will clarify this in the revised version of the paper.
>  - We wish to clarify that we do discuss the 'mixed evidence' results involving comparisons of Responsibility and nearest neighbor in the paper. Moreover, we feel that this further demonstrates the strength of nearest neighbors as a baseline for this work.
>  - Regarding diffusion models, while true that metrics such as SSIM and MSE do not directly measure explainability, we argue that for our use case, as with nearest neighbor, they can be regarded as proxies. A similar image will likely serve as an explanation (“it generated this because the model saw that during training”) at least for human-facing tasks, particularly in lieu of other metrics. We would again greatly appreciate it if you could recommend any other metrics that would more suitably evaluate explainability for diffusion models.
>  - We acknowledge lack of analysis of memory requirements for storing the responsibility arrays. This is dependent on the total number of weights |W| and the total number of training samples |T|, thus requiring a table |W|*|T| in size. We will clarify this in the final paper.
> - Similarly, we acknowledge the lack of discussion of computational overhead during training. Obviously in the batch size of 1 setup this slows down training significantly as one would expect. For the later experiments with larger batch sizes this is essentially a single matrix subtraction (to extract weight deltas) and then iterating over the batch size to update the appropriate responsibility array indices. Thus it is |W|*|b| where |b| is the batch size. In practice, we found this was a marginal slow down.
>  - For determining which layer's weights to leverage for Responsibility, we take inspiration from standard practices in finetuning and transfer learning where it is typical to use the final layer. We will clarify this in the paper.
>  - We wish to point out that the results of different allocation strategies and activation identification methods are given in the appendix.
>  - We really appreciate the various writing and formatting suggestions and will incorporate these into the paper.

---

### Decision · Action_Editor_T4X1 · 2025-09-01

**Recommendation:** Reject

**Audience:**

Yes

**Audience Explanation:**

Yes, explainability is a broad topic and data attribution methods are becoming crucial for LLMs.

**Claims And Evidence:**

No

**Claims Explanation:**

The paper describes a technique to assign, for each prediction, a "responsibility" (influence) to each training point. This is a well known problem in the literature (e.g., influence functions, TracIn, ...). The method stores all individual changes to the weights for each mini-batch, and computes the influence based on the highest (aggregated) change when considering only a few "critical" weights.

All reviewers agree that the method is potentially interesting, the experimental analysis broad, but that the paper has serious drawbacks including (a) lack of theoretical justification, (b) missing related work and baselines, (c) some unconvincing results, (d) lack of serious ablation studies, (e) limited analysis of scaling given by the batch side equal to 1, (f) no consideration of computational issues.

The rebuttal was not very convincing, with too many items (experiments, baselines, theoretical justifications) promised but not shown. As a result, two out of three reviewers remain negative, and I agree with their evaluation. The amount of additions is too large for a minor revision, especially since none of these were shown during the rebuttal. As such, the paper clearly fails in having results "supported by accurate, convincing and clear evidence".